# Saturated Iso-Type Fatty Acids from the Marine Bacterium *Mesoflavibacter zeaxanthinifaciens* with Anti-Trypanosomal Potential

**DOI:** 10.3390/ph17040499

**Published:** 2024-04-13

**Authors:** Dayana Agnes Santos Ferreira, Erica Valadares de Castro Levatti, Lucas Monteiro Santa Cruz, Alan Roberto Costa, Álvaro E. Migotto, Amanda Yaeko Yamada, Carlos Henrique Camargo, Myron Christodoulides, João Henrique G. Lago, Andre Gustavo Tempone

**Affiliations:** 1Pathophysiology Laboratory, Instituto Butantan, Av. Vital Brazil, 1500, Sao Paulo 05503-900, SP, Brazil; dayana.ferreira.esib@esib.butantan.gov.br (D.A.S.F.); ericavclevatti@gmail.com (E.V.d.C.L.); 2Centre of Organic Contaminants, Instituto Adolfo Lutz, Av. Dr. Arnaldo, 355, Sao Paulo 01246-000, SP, Brazil; lucas.cruz@ial.sp.gov.br (L.M.S.C.); alan.costa@ial.sp.gov.br (A.R.C.); 3Centre for Marine Biology, Universidade de São Paulo, Rodovia Doutor Manoel Hipólito do Rego, km. 131,5, Pitangueiras, Sao Sebastiao 11612-109, SP, Brazil; aemigott@usp.br; 4Centre of Bacteriology, Instituto Adolfo Lutz, Av. Dr. Arnaldo, 351, Sao Paulo 01246-000, SP, Brazil; amandayy45@gmail.com (A.Y.Y.); carlos.camargo@ial.sp.gov.br (C.H.C.); 5Molecular Microbiology, School of Clinical and Experimental Sciences, Faculty of Medicine, University of Southampton, Southampton SO16 6YD, UK; M.Christodoulides@soton.ac.uk; 6Centre of Natural Sciences and Humanities, Universidade Federal do ABC, Sao Paulo 09210-580, SP, Brazil

**Keywords:** marine bacteria, *Trypanosoma cruzi*, iso-fatty acids, metabolites, antimicrobial

## Abstract

Chagas disease is a Neglected Tropical Disease with limited and ineffective therapy. In a search for new anti-trypanosomal compounds, we investigated the potential of the metabolites from the bacteria living in the corals and sediments of the southeastern Brazilian coast. Three corals, *Tubastraea coccinea, Mussismilia hispida*, *Madracis decactis*, and sediments yielded 11 bacterial strains that were fully identified by MALDI-ToF/MS or gene sequencing, resulting in six genera—*Vibrio*, *Shewanella*, *Mesoflavibacter*, *Halomonas*, *Bacillus,* and *Alteromonas.* To conduct this study, EtOAc extracts were prepared and tested against *Trypanosoma cruzi.* The crude extracts showed IC_50_ values ranging from 15 to 51 μg/mL against the trypomastigotes. The bacterium *Mesoflavibacter zeaxanthinifaciens* was selected for fractionation, resulting in an active fraction (FII) with IC_50_ values of 17.7 μg/mL and 23.8 μg/mL against the trypomastigotes and amastigotes, respectively, with neither mammalian cytotoxicity nor hemolytic activity. Using an NMR and ESI-HRMS analysis, the FII revealed the presence of unsaturated iso-type fatty acids. Its lethal action was investigated, leading to a protein spectral profile of the parasite altered after treatment. The FII also induced a rapid permeabilization of the plasma membrane of the parasite, leading to cell death. These findings demonstrate that these unsaturated iso-type fatty acids are possible new hits against *T. cruzi.*

## 1. Introduction

Marine microorganisms have developed defense strategies that include the production of different metabolites and enzymes with important applications for humans [1]. These low molecular weight compounds are produced by living organisms to perform biological functions for their survival and their defense [2]. The exploration of marine microbiota is recent, but already shows great promise in both the medical and biotechnology fields [3]. Since the oceans cover most of the Earth’s surface, they host a substantial portion of the biodiversity, which lives under distinct and varied conditions and has evolved through a long period of metabolic adaptations. The exploration of the biodiversity and metabolic chemodiversity of the oceans has resulted in the discovery of thousands of structurally unique bioactive marine natural products [4]. Considering that oceans contain typical microbial abundances up to 10^7^ cells/mL in seawater and 10^10^ cells/cm^3^ in sediments [5], the exploration of marine microorganisms as sources of bioactive marine organisms has led to the discovery of promising new drug candidates [1,2,3,4,6]. Newman and Cragg [7] reported that about 56% of 1881 therapeutic products approved by the US Food and Drug Administration (FDA) between 1981 and 2019 had natural prototypes, especially obtained from marine microorganisms.

Microorganisms might be a potential source of molecules to control parasitic infections such as Chagas disease [8], which is a trypanosomiasis that is considered neglected by the World Health Organization (WHO). This condition affects over six million people worldwide, mainly in Latin America [9]. It is caused by the protozoan parasite *Trypanosoma cruzi*, which is mostly transmitted by the insect *Triatoma* sp. or by the oral contamination of food [10]. In non-endemic countries, it is present due to immigration and disseminated by contaminated food and organ transplantation [11,12]. The parasite can be identified in two clinically relevant forms: (1) trypomastigotes, which are found in blood and extracellular compartments and are considered the infective form, and (2) amastigotes, which are found in the affected tissues during the latent and chronic disease and are the intracellular replicative forms. The current treatment for Chagas disease uses only two highly toxic drugs, benznidazole and nifurtimox. Both drugs are associated with severe adverse effects [10], and only benznidazole is still approved in Brazil. However, a large multicentric study confirmed the low efficacy of benznidazole to eliminate the parasites in chronic patients [13], emphasizing the urgent need for safer and more effective treatments for Chagas disease.

According to the MarinLit database, the marine environment is a source of more than 40,000 compounds from macro- and microorganisms, [14]. In view of the huge chemodiversity present in the microorganisms living in the oceans, numerous biologically active compounds have been described [15]. For example, small molecules isolated from the marine bacterium *Bacillus pumilus*, present in the black coral *Antipathes* sp. of the Pacific coast of Panama, have shown potent activity against *T. cruzi,* with IC_50_ values from 19 to 27 μM [16]. Penidigiamycin-A, an antiprotozoal compound, isolated from the marine bacterium *Paenibacillus* sp. DE2SH, showed anti-*Leishmania* (IC_50_ 7 µM), anti-*T. brucei* (IC_50_ 0.78 µM), and anti-*Plasmodium falciparum* (IC_50_ 9.1 µM) activity [17]. 

The Blue Amazon is the Brazilian oceanic area of the Atlantic that covers 3.6 million square kilometers and presents a huge biodiversity of potential pharmaceutical candidates to fight Neglected Tropical Diseases (NTDs). In this study, we examined the antiparasitic potential of bacterial metabolites isolated from three corals and samples of sediments, which were all collected from the north coast of São Paulo, Brazil (Figure 1, red arrow).

## 2. Results

### 2.1. Collection of Marine Invertebrates 

In our study, we collected three different species of marine invertebrates and two sediment samples by scuba diving in the geographical area of the São Sebastião Channel and Buzios Island (Table 1). The depth of the collection ranged from 5 to 35 m (Table 1), and the collected specimens are shown in Figure 2.

### 2.2. Isolation and Identification of Marine Bacteria

The microbiota associated with the corals were isolated under sterile conditions and stored at −85 °C. The isolation procedure yielded 11 microorganisms, which were selected based on their abundance. The microorganisms were identified with MALDI-ToF/MS (Bruker Daltonics, Bremen, Germany), according to the logarithmic system of intrinsic scoring where the program attributes a degree of confidence (score) to the result (Table 2). 

Considering the limited databank for marine bacteria, the isolates that could not be identified by MALDI-TOF/MS were selected for a partial sequence identification of the 16S rRNA gene. After sequencing, the nucleotide sequences were compared to those deposited in the sequence banks of EzTaxon (https://www.ezbiocloud.net/, accessed 10 August 2023), (https://www.ezbiocloud.net/ accessed on 9 August 2023), SepsiTest (http://www.sepsitest-last.de/en/index; accessed on 10 August 2023), Microbenet (https://microbenet.cdc.gov/, accessed on 10 August 2023), and the BLAST 16S rRNA database. Table 3 shows the respective percentages of similarity based on the sequencing of the partial gene 16S rRNA. The identified organisms are compatible with the marine bacteria *Alteromonas macleodii*, *Shewanella pneumatophori*, *Mesoflavibacter zeaxanthinifaciens*, and *Halomonas aquamarina*.

Thus, the study identified eleven strains belonging to seven different species: *Alteromonas macleodii*, *Vibrio harveyi*, *Vibrio alginolyticus*, *Shewanella pneumatophori*, *Mesoflavibacter zeaxanthinifaciens*, and *Halomonas aquamarina*. 

### 2.3. Extraction of Microbial Metabolites and Evaluation of the 50% Inhibitory Concentration (IC_50_) for T. cruzi Trypomastigotes

To evaluate the anti-*T. cruzi* potential of microbial metabolites, crude organic extracts were obtained after cultivation in a Marine Agar medium in 140 mm × 15 mm Petri dishes. A total of seven plates were used, with cultivation for 120 h for each species. The organic extracts of the 11 strains isolated from the marine and sediment samples were obtained and ranged from 2 to 11 mg (Table 4).

The 50% inhibitory concentration (IC_50_) of the organic extracts were quantified against trypomastigote forms of *T. cruzi*, and all 11 extracts had antiparasitic activity, with IC_50_ values ranging from 8 to 60 μg/mL. The least active extract was from the *B. megaterium* (60 µg/mL), which was isolated from the sediments of the São Sebastião Channel (Table 4). The most active extracts were from that *Vibrio* spp. that was isolated from different corals and marine sediments, with IC_50_ values ranging from 8 to 51 μg/mL, as well as the *S. pneumatophori* (15 μg/mL), *M. zeaxanthinifaciens* (18 μg/mL), and *H. aquamarina* (15 μg/mL). 

Based on the dose–response sigmoidal curves (Appendix A), six microbial extracts killed 100% of parasites at the highest tested concentration and were ranked as follows: TC 2.2 (*V. alginolyticus*), MH 3.3 (*V. alginolyticus*), SCSB 6.0.2.2 (*M. zeaxanthinifaciens*), SCSB 6.2 (*V. harveyi*)*,* SIBUZ 7 *(V. harveyi*), and SIBUZ 7.2.2. (*H. aquamarina*). In addition, with an assessment of the cellular viability with the resazurin reagent that measures mitochondrial oxidative activity, it was possible to verify that all the extracts showed a trypanocidal activity. 

### 2.4. Fractionation of Mesoflavibacter zeaxanthinifaciens Extract

Several factors were considered in choosing the organism for further fractionation studies. We selected *M. zeaxanthinifaciens* on account of its potency, the amount of the EtOAc extract obtained, and the fact that no previous fractionation study has been conducted with this microorganism to our knowledge. Thus, chromatographic fractionation of *M. zeaxanthinifaciens* produced four fractions—F0, FI, FII, and FIII.

After evaluation of the biological activity of the fractions against trypomastigotes, fraction II (FII) eliminated 100% of the parasite after 24 h at a tested dose of 100 µg/mL. Next, this fraction was analyzed by NMR and by ESI-HRMS (Figure 3) to identify the main bioactive compounds.

Initially, the NMR ^1^H spectrum indicated the predominance of fatty material due to the presence of an intense singlet at δ 1.2. However, the signal referring to the termination of the carbon chain, commonly observed as one triplet at δ 0.8, was detected as one doublet (J = 6.5 Hz), suggesting the presence of iso-fatty acids (Appendix A). To identify these compounds, this fraction was analyzed with an ESI-HRMS in negative mode (Figure 3).

According to the literature, the presence of iso-type fatty acids has been described previously in the study species [18]. Thus, the occurrence of such fatty acids with variations in the extension of the lateral chain was proposed based on the obtained results of the ESI-HRMS data. 

As shown in Figure 3, four main [M–H]- ions at *m*/*z* 297.2790, 311.2954, 325.3104, and 339.260 were observed, which were compatible with the molecular formulas C_19_H_37_O_2_- (calculated 297.2794), C_20_H_39_O_2_- (calculated 311.2950), C_21_H_41_O_2_- (calculated 325.3106), and C_22_H_43_O_2_- (calculated 339.3263), all with a degree of unsaturation, i.e., a carbonyl, therefore containing saturated side chains. This analysis allowed the identification of a homologous series of C_19_, C_20_, C_21_, and C_22_ iso-type fatty acids (compounds **1**–**4**), as shown in Figure 4.

### 2.5. Antitrypanosomal Activity of Fraction II of Mesoflavibacter zeaxanthinifaciens EtOAc Extract (FII) 

The potency of fraction II of the *M. zeaxanthinifaciens* EtOAc extract was determined against *T. cruzi* trypomastigotes and intracellular amastigotes. This fraction showed potent activity against the trypomastigotes with an IC_50_ value of 17.7 μg/mL after 48 h of incubation. The intracellular amastigotes were also eliminated with an IC_50_ value of 23.8 μg/mL. Furthermore, an assessment of the cytotoxicity of fraction II in NCTC clone 929 cells revealed no toxicity at the highest tested concentration of 200 μg/mL (Table 5).

### 2.6. Hemolytic Activity of Fraction II of Mesoflavibacter zeaxanthinifaciens EtOAc Extract

The hemolytic activity of fraction II was evaluated on murine erythrocytes after incubation for 2 h, using water as a positive control (Figure 5). No hemolytic activity could be detected up to 100 μg/mL, and <10% of the erythrocytes were hemolyzed at a concentration of 200 μg/mL, compared to the untreated cells (the negative control). Water alone produced 100% hemolysis (the positive control).

### 2.7. Protein Profile of T. cruzi after Treatment with Fraction II of Mesoflavibacter zeaxanthinifaciens EtOAc Extract (FII)

The protein profile of the trypomastigotes was evaluated with the MALDI-TOF/MS (Bruker Daltonics, Bremen, Germany) after treatment of the parasites with fraction II of the *M. zeaxanthinifaciens* EtOAc extract (FII) at two concentrations (20 and 50 μg/mL). The parasites were also treated with benznidazole (BZN-40 μM) for comparison. Based on the spectral data of the untreated parasites, significant changes in the protein profile of fraction II-treated parasites were observed (Figure 6).

Significant spectral modifications were observed at peaks *m*/*z* 4117, with decreased intensities in the group treated with FII at 50 µg/mL, when compared to the untreated parasites and those treated with benznidazole (Appendix A). Similarly, the areas of the peaks at *m*/*z* 2690, 2780, and 30,975 were significantly decreased for the FII-treated parasites (50 μg/mL), as well as those above *m*/*z* 10,000. However, an increased peak intensity was found with the FII-treated group at 50 µg/mL and *m*/*z* 8075, and a similar effect was also detected in the benznidazole-treated group. The effect of the FII treatment in the *T. cruzi* was clearly dose-dependent, and the alterations of the protein spectra were like those found for benznidazole, the standard drug.

### 2.8. Plasma Membrane Permeability of Trypomastigotes Treated with Fraction II of Mesoflavibacter zeaxanthinifaciens EtOAc Extract

The plasma membrane permeability of the trypomastigotes was analyzed in the presence of fraction II of the *M. zeaxanthinifaciens* EtOAc extract (FII) at the IC_50_ value (17.7 µg/mL), using the fluorescent probe SYTOX Green. After 10 min of incubation, the FII induced a rapid permeabilization of the membrane, with a significant increase in fluorescence levels when compared to the untreated control (Figure 7). Triton X-100 (0.5% *v*/*v*) was used as a positive control and showed the highest levels. After 60 min of incubation, the fluorescence levels of the fraction II-treated group were like those that were treated with Triton X-100.

Using fluorescence digital microscopy, it was also possible to corroborate the spectrofluorimetric data by taking images of the parasites after the incubation with FII (Figure 8). In the FII-treated group, the parasites showed an altered morphology, with a round shape and an intense green fluorescence in the cytoplasm, caused by the penetration of the vital SYTOX Green dye. The untreated group showed no fluorescence, and the parasites demonstrated a normal elongated morphology (Figure 8). The positive control with Triton X-100 demonstrated an intense alteration of the parasites’ morphology with an elevated green fluorescence.

## 3. Discussion

Chagas disease is a serious public health problem in 21 developing countries [9]. In Brazil, only one drug, benznidazole, is approved for treatment, and it has a limited efficacy and high toxicity. In 2015, Morillo and co-workers published the most complete multicenter randomized clinical study with benznidazole, which involved 2854 patients with the Chagas cardiomyopathy [13]. Their results showed that benznidazole therapy significantly reduced parasitemia but showed no efficacy to reduce cardiac damage over the 5-year follow-up of the study [13]. There is a consensus in the literature around the urgent need for new therapeutic candidates for Chagas disease. 

Considering the huge chemodiversity found in marine microorganisms, we investigated for the first time, to our knowledge, the anti-*T. cruzi* potential of the metabolites produced by bacteria that had been previously isolated from invertebrates and sediments from the Sao Paulo north coast in Brazil. Among the active extracts, we observed that metabolites of the bacterium *Vibrio harveyi* were amongst the most potent. This strain was isolated from four sources, one from the coral *M. hispida* and two from the sediments from the São Sebastião Channel and the Buzios Island at depths of 35 and 13 m, respectively. The members of the Vibrionaceae family live freely in tropical marine waters or in the microbiota of marine animals [19]. The genus *Vibrio* was the most common found in this study, covering seven of the eleven identified strains, and it has been recognized as part of the microbiome of several invertebrates. It is a holobiont that participates in nitrogen fixation and organic matter recycling and acts as a food resource and for chitin decomposition amongst other ecological roles, and some species are pathogenic to animals and humans [20].

The Vibrionaceae family has yielded 93 metabolites with different biological activities, including antifungal, antibacterial, and anticancer. Most have been isolated from three species, namely, *V. parahaemolyticus, V. anguillarum*, and *V. vulnificus*. However, only one compound, prodigiosin, showed antiprotozoal activity against *Plasmodium* spp. [21]. 

To our knowledge, our study has demonstrated the anti-trypanosomal activity of metabolites from *Vibrio* spp. for the first time. It was noteworthy that the same species, collected from different locations, resulted in different potencies against the parasites. This was confirmed with *V. harveyi*, which produced metabolites with IC_50_ values against *T. cruzi* ranging from 8 to 51 µg/mL. These differences in the metabolic profile of genetically similar strains have been reported for *Streptomyces griseus*, an exemplar of the most prolific genus in the production of bioactive compounds [22]. The species closely associated by 16S rRNA maintain a common set of chemical compounds. However, despite the identical sequences of this gene, a noticeable difference was found in the accessory set of accumulated metabolites, which are unique for each bacterium [21]. Sottorff and colleagues suggested that, based on the phylogenetic proximity and the similarity of the metabolites, both strains of *Streptomyces* spp. had a common origin that underwent posterior specialization as a function of their habitat [21]. In our studies, the *Vibrio* spp. strains were isolated from completely different environments, which included deep sediments (close to the coast), sediments from the shallow waters of an island (approximately 20 miles from the coast), and three coral species. It is possible that the different microbiomes may have contributed to the process of the specialization of their metabolites.

Another potent extract with anti-*T. cruzi* activity was produced from *Shewanella pneumatophori*, isolated from marine sediments of the Buzios Island. This bacterium belongs to a genus generally found in extreme temperatures and it has shown probiotic features as a positive modulator of fish health [20]. The antimicrobial ability of a biosurfactant produced by *Shewanella algae* strain B12 has also been revealed [21]. Our study demonstrated, for the first time in the literature, the anti-*T. cruzi* activity of metabolites from this genus. 

The genus *Bacillus* has already been reported to exhibit antimicrobial activities e.g., *B. subtilis* [23], and antiparasitic activities e.g., *B. pulmilus* against *T. cruzi* [16]. In our study, *Bacillus megaterium*, isolated from deep sediments of the São Sebastião Canal at a depth of 35 m, showed anti-*T. cruzi* activity, but with the lowest potency amongst all tested extracts. This species is generally found in terrestrial environments and has the capacity to solubilize natural phosphates in soil and are therefore of great interest for the pharmaceutical bioproducts industry [24]. The species can also provide antibacterial molecules [25]. 

Our study is also the first to describe the highly potent activity of *Halomonas aquamarina* metabolites against *T. cruzi* The *Halomonas* genus has a wide geographical distribution and promising antimicrobial and antitumor biological activities [26,27]. *Halomonas* is a common genus of halophilic bacteria, which are inhabitants of hypersaline environments. These extremophiles have special abilities to produce extremozymes and other bioactive molecules as potential antibiotics [28]. 

Another potent microbial extract against *T. cruzi* was obtained from *M. zeaxanthinifaciens*, a bacterium isolated from the marine sediments of Buzios Island. This is an aerobic Gram-negative, rod-shaped, halo- and mesophilic bacterium, first isolated from a seawater sample collected from the Pacific coastline of Japan [18]. This species can produce the carotenoid pigment zeaxanthin (3,3′-dihydroxy-β-carotene), which is a yellow lipophilic molecule useful for the pigmentation of food products and cosmetics [29]. Zeaxanthin is also a strong antioxidant with anticancer properties, and it has significant potential for pharmaceutical use, especially for preventing age-related macular degeneration and other eye-related diseases [29]. Some *Flavobacteria* and other *Mesoflavibacter* have also been reported to produce zeaxanthin, amongst other metabolites, and might be useful for the biotechnological production of this important compound [30,31]. 

*M. zeaxanthinnifaciens* also produces exopolysaccharides, and, when associated with nanoparticles, these macromolecules show antibacterial and antibiofilm properties against pathogenic microorganisms such as *Bacillus subtilis* and the methicillin-resistant *Staphylococcus aureus* [32]. Our study is the first to explore the antiparasitic capacity of compounds from this strain, which was also identified in Brazil for the first time. 

Considering the antiparasitic potential of these metabolites, we conducted an OSMAC (One-Strain Many Compounds) study, a promising approach to activate cryptic genes and to enhance the production of bioactive compounds. Using solid and liquid media for cultivation, our results showed that cultivation in a liquid medium with rotation resulted in a higher mass of microbial metabolites. However, the potency of these anti-*T. cruzi* metabolites decreased 8-fold when compared to those produced by bacteria grown on a solid medium (Marine Agar). English and co-workers used a similar OSMAC approach to test the ability of *Streptomyces* sp. to produce antibiotics against *Staphylococcus aureus*. No antimicrobial activity was detected in metabolites isolated from bacteria cultured in a liquid medium, but only on an agar medium [33]. Similarly, Guo and co-workers also observed that the marine fungus *Penicillium* sp. F23-2 produced different chemical compounds when cultivated on a solid medium, leading to the isolation of five new analogues of ambuic acid (penicyclones A-E) with antibacterial activity [34].

We selected the metabolites of *M. zeaxanthinifaciens* for the bio-guided fractionation. The rationale for their selection was based on the novelty of their antiparasitic potential, and, amongst the isolated strains, this species showed a good relationship between the IC_50_ value and the yield of metabolites (mass). Additionally, to the best of our knowledge, we demonstrated the isolation of *M. zeaxanthinifaciens* in the Atlantic Ocean for the first time, as it has previously only been described in the Japanese Pacific Ocean [18]. Using a solid-phase extraction, we fractionated a crude EtOAc extract from *M. zeaxanthinifaciens,* which resulted in fraction FII that showed trypanocidal activity like the crude extract, as confirmed by the lack of mitochondrial activity detected by the resazurin assay. Usually, fractionated samples are more active than the original crude extract, because the fractionation procedure itself concentrates the active compounds. In our study, FII was not the most abundant fraction, corresponding to 2.9% of the total mass of the crude extract, but this fact cannot explain the similarity between the IC_50_ values. Considering that synergism between compounds in crude extracts has been widely reported [35], the separation of compounds in the FII from those in the F0, FI, and FIII may have contributed to reducing the potency against *T. cruzi* parasites.

The studies with the intracellular amastigotes revealed a potent activity of FII against the clinically relevant form of the parasite. Despite the importance of the trypomastigotes during the acute phase of the disease, amastigotes are the major parasitic form living inside cells and contributing to the prevalence of the disease over decades [36].

It was noteworthy that FII eliminated both forms of the parasite without affecting the mammalian cells as demonstrated by the lack of cytotoxicity against fibroblasts, up to a tested concentration of 200 μg/mL. Additionally, FII was non-hemolytic at the same high concentration. These studies demonstrated that FII presented selective compounds against *T. cruzi*, suggesting a promising biological activity for the drug discovery exploration. 

Using ^1^H NMR and ESI-HRMS analysis, four iso-fatty acids (1: C_19_H_37_O_2_, 2: C_20_H_39_O_2_, 3: C_21_H_41_O_2_, and 4: C_22_H_43_O_2_) were identified as the main compounds in FII. Our study corroborated the known predominance of the branched saturated fatty acids, in addition to branched monounsaturated and branched hydroxyacids in specimens of the family *Flavobacteriaceae* [18,37]. Studies on the antiparasitic activity of fatty acids are limited. Previous reports have shown that acetylenic fatty acids from *Porcelia macrocarpa* affected *T. cruzi* by causing a disturbance in the parasite plasma membrane [38]. Other fatty acids have been reported with antimalarial, antimycobacterial, and antifungal properties. It is possible that these can induce pore formation in bacterial membranes and alter cell permeability, activating signaling pathways that lead to microbial death [38]. 

Considering the promising results of fraction II of the *M. zeaxanthinifaciens* EtOAc extract against *T. cruzi,* we investigated the mechanism of action in the parasite. Using MALDI-ToF/MS, our studies demonstrated that FII induced significant dose-dependent alterations of the protein profile of trypomastigotes after treatment, reducing the intensities of protein signals when compared to the untreated parasites and like the standard drug benznidazole. This effect has also been shown for 6-brome-2′-de-N-methylaplysinopsin, a marine alkaloid isolated from the coral *Tubastrea tagusensis* with potent activity against *T. cruzi* [39]. The *T. cruzi* plasma membrane acts as a physical barrier between the external environment and inner cell organelles, allowing the exchange of molecules and the maintenance of the cell potential and plays a vital role in the shape of the cell [40]. In our study, the plasma membrane of trypomastigotes was rapidly permeabilized in the presence of FII, and this was accompanied by the loss of the elongated shape of trypomastigotes to an amastigote-like form. It is known that permeabilization of the plasma membrane can contribute to the leakage of intracellular components leading to cell death. For example, amphotericin B, an antifungal approved drug used for the treatment of leishmaniasis, alters the permeability of the *Leishmania* parasite by inducing membrane pore formation [41]. It is possible that a similar mechanism occurs in *T. cruzi* treated with FII.

## 4. Materials and Methods

### 4.1. Mice

The BALB/c mice were obtained from the animal breeding facility at the Instituto Adolfo Lutz, Sao Paulo, Brazil. The animals were maintained in sterilized cages under a controlled environment and received water and food ad libitum. All procedures were approved by the Animal Care and Use Committee from the Instituto Adolfo Lutz—Secretary of Health of Sao Paulo State (Project number CEUA 05/2018) in agreement with the Guide for the Care and Use of Laboratory Animals from the National Academy of Sciences. 

### 4.2. Parasites and Mammalian Cell Maintenance

The macrophages were collected from the peritoneal cavity of the BALB/c mice by washing the cavities with an RPMI-1640 medium (Sigma-Aldrich, St. Louis, MO, USA) supplemented with 10% (*v*/*v*) of fetal calf serum (FCS, Sigma-Aldrich, USA) and were maintained at 37 °C in a 5% (*v*/*v*) CO_2_ humidified incubator. Trypomastigotes of the *T. cruzi* (Y strain) were maintained in *Rhesus* monkey kidney cells (LLC-MK2—ATCC CCL 7), cultivated in an RPMI-1640 medium supplemented with 2% (*v*/*v*) of fetal calf serum at 37 °C in a 5% (*v*/*v*) CO_2_ humidified incubator. The murine conjunctival cells (NCTC clone 929, ATCC) were maintained in the RPMI-1640 supplemented with 10% (*v*/*v*) of FCS at 37 °C in a 5% (*v*/*v*) CO_2_ humidified incubator. 

### 4.3. Analytical Methods

The ^1^H NMR spectra (500 MHz) were recorded on a Varian INOVA 500 spectrometer using CD3OD (Sigma-Aldrich) as the solvent and internal standard. The ESI-HRMS spectra were obtained with an electrospray ionization in the positive ion mode on a Bruker Daltonics MicroTOF QII spectrometer. The MALDI-TOF/MS data were analyzed on a Bruker MicroFlex spectrometer at a 20 kV accelerating voltage in the positive mode (500 laser shots). The signals were collected in a range between *m*/*z* 2000 and 20,000 with the AutoXecute tool 3.0 (Bruker Daltonics, Bremen, Germany). 

### 4.4. Collection of Marine Corals and Sediments

The collections were made by scuba diving in the São Sebastião region at CEBIMar-USP at depths ranging from 5 to 35 m. The marine invertebrates were identified by Dr. Álvaro E. Migotto (ICMBio MMA 10186-2). The sea sediments were collected with the help of a dredge in the São Sebastião Channel (at a depth of 35 m) (S 23°49′40.7″; W045°24′44.7″), as well as on Buzios Island by scuba diving (13 m deep) (23°48′05.2″; W 45°08′32.2″).

### 4.5. Isolation and Cultivation of Microorganisms

After collection, the samples were immediately processed at the Centro de Biologia Marinha (CEBIMar), Universidade de São Paulo, under sterile conditions, and the contents of the corals and sediment samples were inoculated onto 90 mm × 15 mm Petri dishes in a BOD incubator (SolidSteel, Piracicaba, Brazil) at 25 °C, using an agar medium rich in nutrients and specific for heterotrophic marine bacteria (Marine Agar, Difco, Tucker, GA, USA). Several isolates were obtained from this material. The isolates were stored at −85 °C in a liquid medium specific to heterotrophic marine bacteria (Marine Broth, Difco, Tucker, GA, USA) containing 15% (*v*/*v*) of DMSO (Merck, Rahway, NJ, USA).

### 4.6. Bacteria Identification with Mass Spectrometry (MALDI-ToF/MS)

MALDI-TOF/MS (Matrix Associated Laser Desorption–Ionization—Time of Flight) analysis was used to identify the microorganisms. The samples were extracted in 1.5 mL tubes or directly onto the plate. For the tube extraction, up to five colonies were suspended in 300 µL of ultrapure water (Milli-Q) and mixed by vortex. Next, 900 µL of EtOH was added, and the vortex was mixed and centrifuged (Eppendorf, Hauppauge, NY, USA) at maximum speed for 1 min. After elimination of the EtOH, 50 µL of 70% (*v*/*v*) of formic acid and 50 µL of acetonitrile (100%) were added to the pellet. The tube was centrifuged for 2 min and the supernatant was dispensed in duplicates (1 µL) into a 96-well plate for the MALDI-TOF/MS and air-dried. The matrix, α-cyano-4-hydroxy-cinnamic acid (HCCA) (Bruker Daltonics, Bremen, Germany), was prepared at a concentration of 50 mg/mL in 50% (*v*/*v*) of acetonitrile and 50% (*v*/*v*) of water with 2.5% (*v*/*v*) of trifluoroacetic acid, and 1 µL of the matrix was added to the dried sample. In the direct plate extraction, a few colonies were transferred to the 96-well plate and covered with the matrix as described above [42]. A DH5-α protein extract from *Escherichia coli* (Bruker Daltonics, Bremen, Germany) was added to the plate as an external control. The analyses were performed on the MALDI-TOF/MS Microflex mass spectrometer (Bruker Daltonics, Bremen, Germany) with a nitrogen laser (337 nm) operating in linear mode with a delayed extraction (260 ns) at a 20 kV accelerating voltage. Each spectrum was automatically collected in the positive ion mode as an average of 500 laser shots (50 laser shots at 10 different point positions). A mass range between 2000 and 20,000 *m*/*z* (mass-to-charge ratio) was selected to collect the signals with the Auto Xecute tool of the flexcontrol acquisition software (Version 2.4; Bruker Daltonics, Bremen, Germany). Only the peaks with a signal-to-noise ratio were considered. The software utilizes a logarithmically transformed similarity score that is computed by evaluating several factors. Log(score) values above 2 indicate the species-level identification of bacteria. Scores between 1.7 and 2.0 suggest the identification of microorganisms at the genus level. Scores below 1.7 signify that the spectrum cannot be effectively identified using the MALDI Biotyper method [42].

### 4.7. Bacterial Identification by Sequencing of the Partial 16S rRNA Gene

Some strains were not identified by the MALDI- ToF/MS method and were selected for genetic sequencing. Genomic DNA from the bacterial culture was isolated using the Wizard^®^ Genomic DNA Purification Kit, with some modifications. In this modified protocol, we increased the incubation period with Proteinase K (2 h) and used RNase (10 µL, 0.5 μg/μL). The partial amplification of the 16S rRNA gene was performed with the GoTaq Master Mix kit (Promega, Madison, WI, USA) and universal primers for *Eubacteria* [43]. The PCR products were purified using the ExoSAP-IT (Thermo Fisher Scientific, Waltham, MA, USA), and the sequencing reactions were prepared using the BigDye^®^ Terminator v3.1 Cycle Sequencing Kit (Thermo Fisher Scientific). A precipitation reaction with ethanol–sodium acetate was performed, and the products were sequenced using the ABI 3730 DNA Analyzer, a 48-capillary analysis system (Applied Biosystems, Waltham, MA, USA). The sequences were analyzed using BioNumerics v.8.0 software (Applied Maths, BioMerieux, Durham, NC, USA) in comparison with the sequences present in the Ezbiocloud and Microbenet databases for the species database and in the NCBI (http://www.ncbi.nlm.nih.gov/, accessed on 15 September 2023), using the Basic Local Alignment Search Tool (BLAST), and the 16S ribosomal RNA sequences databases.

### 4.8. Extraction of Metabolites from Marine Bacteria Grown on Agar and in Broth Medium

#### 4.8.1. Agar

The bacterial strains were seeded in Petri dishes (140 mm × 15 mm) and grown for 120 h at 25 °C. The colonies were scraped with a cell scraper (Corning, Corning, NY, USA) and diluted in ultrapure water (200 mL for every 20 plates) in a glass vial. After vortex mixing for 2 min and sonication in an ultrasonic bath for 10 min, 200 mL of EtOAc (JT Baker, Segrate, Italy) was added, and the solution was transferred to the ultrasonic bath for 40 min. After partitioning, the organic phase was filtered, and the solvent was removed under a reduced pressure at 40 °C to afford the crude EtOAc extract.

#### 4.8.2. Broth

The colonies grown on Marine Agar (Difco, Franklin Lakes, NJ, USA) were transferred to 20 mL of a liquid medium Marine Broth and cultured for 24 h in a BOD incubator (SolidSteel, Piracicaba, Brazil) at 25 °C. The optical density of this culture was measured and adjusted to 0.5 on the McFarland scale, i.e., 1.5 × 10^8^ Colony Forming Units (CFU)/mL. The culture was kept at 25 °C in an incubator (SolidSteel, Brazil) with an orbital shaker or static for a total of 120 h. After this period, the cultures were transferred to 50 mL plastic conical tubes and centrifuged at 4000× *g* for 20 min. The cell precipitate was separated from the supernatant and concentrated in 100 mL of ultrapure water (Milli-Q). After vortex mixing for 2 min and sonication in an ultrasonic bath for 10 min, 100 mL of EtOAc was added to this aqueous phase. The extraction procedure was then followed.

### 4.9. Bio-Guided Fractionation of Mesoflavibacter zeaxanthinifaciens Organic Extract

The fractionation of the EtOAc extract of the *M. zeaxanthinifaciens* (500 mg) was performed using a solid phase extraction cartridge (SPE Bond Elute C18—Agilent, Santa Clara, CA, USA). The cartridge was activated with MeOH, conditioned with MeOH:H_2_O (9:1 *v*/*v*), and, after application of the crude EtOAc extract (500 mg), four fractions (F0—158.2 mg, FI—11.9 mg, FII—10.2, and FIII—161.6 mg) were collected using different mixtures of MeOH:H_2_O (1:9, 3:7, 6:4 *v*/*v*, and pure MeOH, respectively) as eluents. The fractions were dried over a reduced pressure and subjected to evaluation of the anti-*T. cruzi* activity.

### 4.10. Evaluation of the Anti-Trypanosoma cruzi Activity

The LLC-MK2-derived trypomastigotes were seeded (1 × 10^6^ cells/well) in 96-well plates and incubated with the extracts/compounds serially diluted 2-fold (200 to 1.5 µg/mL) for 24 h in an RPMI 1640 medium at 37 °C in a 5% (*v*/*v*) CO_2_ humidified incubator. Subsequently, resazurin (0.011% *w*/*v* in PBS) was added to check for parasite viability for 24 h. The optical density was determined in the FilterMax F5 (Molecular Devices, San Jose, CA, USA) at λ 570 nm. Benznidazole was used as the standard drug [44]. 

For the anti-amastigote assay, mouse peritoneal macrophages (1 × 10^5^ cells/well) were seeded in a 16-well chamber slide (NUNC, Thermo Fisher Scientific) and infected with trypomastigotes (10:1 parasite—macrophage ratio). After 2 h, the extracts/compounds were diluted in different concentrations (200 to 1.5 µg/mL) and incubated with infected macrophages for 48 h at 37 °C in a 5% (*v*/*v*) CO_2_ humidified incubator. Finally, the slides were fixed with methanol (100%), stained with Giemsa, and observed under a light microscope (EVOS M5000, Termo, Long Beach, CA, USA) with digital image acquirement. The 50% inhibitory concentration (IC_50_) values were determined by the infection index. Benznidazole was used as the standard drug [39,44].

### 4.11. Cytotoxicity against Mammalian Cells

The fibroblast NCTC cells (clone 929) (6 × 10^4^ cells/well) were seeded in 96-well plates and incubated with the extracts/compounds (1.5 to 200 µg/mL) for 48 h at 37 °C in a 5% (*v*/*v*) CO_2_ humidified incubator. The 50% cytotoxic concentration (CC_50_) was determined by the MTT colorimetric assay, as described previously [39].

### 4.12. Hemolytic Activity

The whole blood was collected from the male BALB/c (25 g) mice after euthanasia. The erythrocytes were collected by centrifugation and used to prepare a 3% (*v*/*v*) solution in PBS. In a round-bottomed titration microplate, the fraction obtained from the active extract of the *M. zeaxanthinifaciens* (FII) was dispensed in serial dilutions of concentrations from 200 to 1.5 µg/mL. Then the suspension of the erythrocytes (100 µL/well) was incubated for 2 h at 24 °C and then incubated for 2 h at 25 °C in a BOD incubator (Solidsteel, Piracicaba, Brazil). The supernatant was collected, and the optical density was determined at λ 570 nm (FilterMax F5 Multi-Mode Microplate Reader, Molecular Devices, San Jose, CA, USA). The maximum hemolysis was obtained using hemocytes suspended in ultrapure distilled water (Milli-Q) as the positive control and untreated hemocytes as the negative control [45]. 

### 4.13. Evaluation of T. cruzi Trypomastigotes Protein Profile

The trypomastigotes (1 × 10^7^/well) were treated with the FII fraction from the *M. zeaxanthinifaciens* (25 µg/mL and 50 µg/mL) or benznidazole (40 µM) for 24 h in an RPMI medium. After this period, the sample suspensions from the expanded cultures were centrifuged, the supernatant removed, and the precipitate was washed twice in Milli-Q water. The precipitate was suspended in 300 µL of Milli-Q water before adding 900 µL of 70% EtOH (*v*/*v*). After further centrifugation, 20 µL of 70% formic acid and 20 µL of acetonitrile were added to the precipitate, and the solution was vortexed and centrifuged. Each centrifugation step was performed at 10,000× *g* for 10 min at room temperature. Untreated parasites were used as the control.

The supernatant was dispensed (3 µL) in duplicates into a 96-well steel plate for the MALDI-ToF/MS (Bruker Daltonics, Bremen, Germany) and dried at room temperature. The matrix, α-cyano-4- hydroxy-cinnamic acid (HCCA) (Bruker Daltonics, Bremen, Germany), was prepared at a concentration of 50 mg/mL in 50% of acetonitrile and 50% of H_2_O with 2.5% of TFA, and was added (1 µL) onto the plate. A DH5-alpha protein extract from *Escherichia coli* (Bruker Daltonics, Bremen, Germany) was added to the plate as the external control. The analyses were performed on a Bruker Daltonics Microflex MALDI-ToF/MS mass spectrometer with a nitrogen laser (337 nm) operating in linear mode with delayed extraction (260 ns) at a 20 kV accelerating voltage. Each spectrum was automatically collected in the positive ion mode as an average of 500 laser shots (50 laser shots at 10 different point positions). A mass range between 3000 and 20,000 *m*/*z* (mass-to-charge ratio) was selected to collect the signals with the Auto Xecute tool of the FlexControl acquisition software (Version 2.4; Bruker Daltonics, Bremen, Germany). Only the peaks with a signal-to-noise ratio were considered [45]. 

### 4.14. Evaluation of Plasma Membrane Permeability

Sytox^®^ Green, a fluorescent nucleic acid marker, impermeable to viable cells, was used to evaluate the possible changes in the permeability of the plasma membrane. T. *cruzi* trypomastigotes were added to 96-well black microtiter plates (2 × 10^6^ parasites/well) and incubated with Sytox^®^ Green (1 µM) in an HBSS medium supplemented with NaHCO_3_ (4.2 mM) and D-glucose (10 mM). Subsequently, a basal reading of the plate was done, and the FII was added (t = 0) at the IC_50_ value. The plate was incubated at 24 °C for 15 min and the fluorescence signal was monitored every 20 min for 180 min. The measurements were made in a spectrofluorometer (FilterMax F5, Molecular Devices, Molecular Devices, San Jose, CA USA) with excitation filters of λ 485 nm and emission filters of λ 535 nm. The maximum permeabilization was obtained in the presence of 0.5% (*v*/*v*) of Triton X-100, and untreated parasites were used as the negative control (100% viability, integral membrane). Digital images of the trypomastigotes after treatment with the FII fraction from the *M. zeaxanthinifaciens* were obtained with a digital fluorescence microscope (EVOS M 5000, Thermo, USA) using the SYTOX Green [46].

### 4.15. Statistical Analysis

The IC_50_ and CC_50_ values were calculated from sigmoidal dose–response curves. Unless otherwise stated, the data reported were the mean ± standard error of at least two independent experiments performed with duplicate samples. For the hemolytic activity, a one-way ANOVA with Tukey’s multiple comparison test was applied for significance (*p* value < 0.05) using the GraphPad Prism 6.0 software. The samples were tested in duplicate/triplicate, and the experiments were repeated at least twice.

## 5. Conclusions

Our study demonstrates, for the first time, the anti-trypanosomal potential of microbial metabolites from six genera of marine bacteria collected from Blue Amazon corals and sediments. Saturated iso-type fatty acids were identified as the main compounds from a bioactive fraction II of the *M. zeaxanthinifaciens* EtOAc extract (FII). This fraction had a trypanocidal effect against trypomastigotes and the intracellular amastigotes of the parasite, which was due to alteration of the parasite membrane permeability. Moreover, the FII was non-toxic to mammalian cells. We believe that these marine bacteria could be considered as promising sources for the selection of new hit compounds for treating Chagas disease.

## Figures and Tables

**Figure 1 pharmaceuticals-17-00499-f001:**
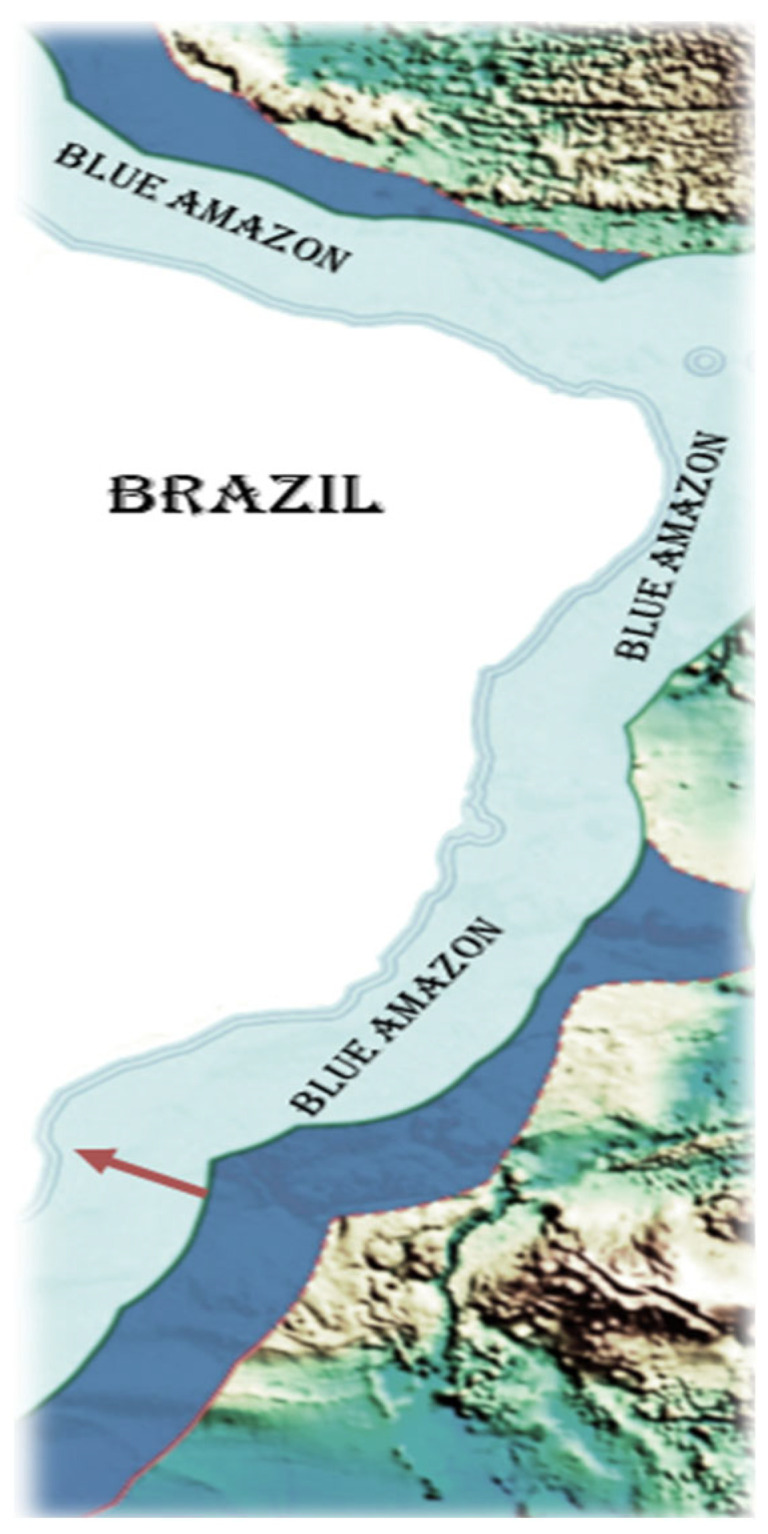
Map comprising the Blue Amazon area of Brazil. The red arrow indicates the approximate locations where the samples were collected by scuba diving: (S 23°48′05.2″; W 45°08′32.2″) and (S 23°49′40.7″; W 45°24′44.7″). Source: the author.

**Figure 2 pharmaceuticals-17-00499-f002:**
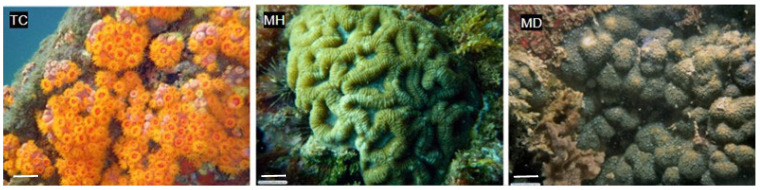
Marine invertebrates collected in the São Sebastião Channel and Buzios Island, São Paulo, Brazil. TC: *Tubastraea coccinea*; MH: *Mussismilia hispida*; and MD: *Madracis decactis*. Photos from Álvaro E. Migotto and Marcelo Visentini Kitahara. Available at the Cifonauta Image Bank (University of São Paulo, http://cifonauta.cebimar.usp.br/, accessed on 10 July 2023). Scale bar (white, left bottom) TC: 20 cm; MH: 25 cm; MD 25 cm.

**Figure 3 pharmaceuticals-17-00499-f003:**
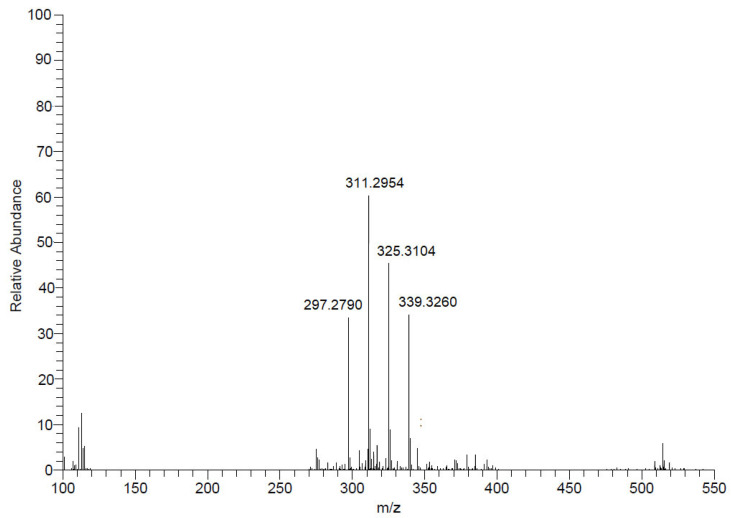
High-resolution mass spectra of fraction II of the *M. zeaxanthinifaciens* EtOAc extract (FII) obtained in a spectrometer (Bruker Daltonics MicroTOF QII), using an electrospray ionization source, operating in negative mode via direct injection.

**Figure 4 pharmaceuticals-17-00499-f004:**
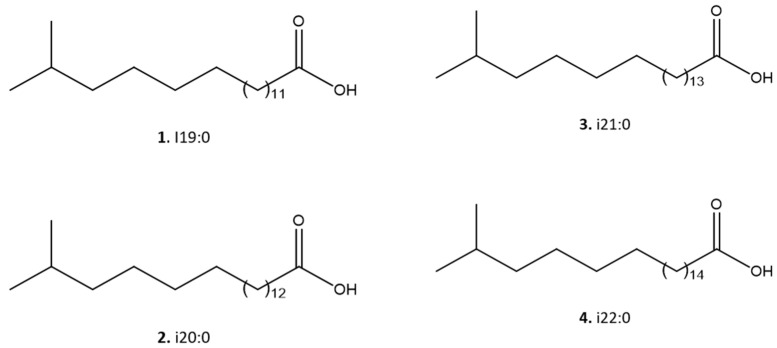
Molecular structures of the iso-chain fatty acids **1**–**4** identified in the bioactive fraction II of the *M. zeaxanthinifaciens* EtOAc extract.

**Figure 5 pharmaceuticals-17-00499-f005:**
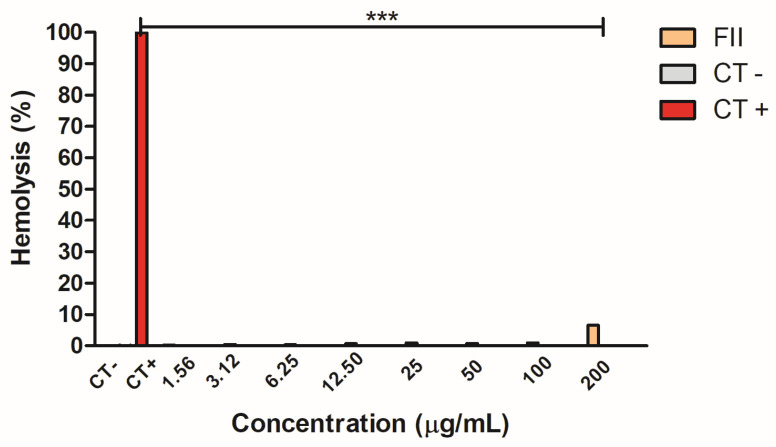
Hemolytic activity of fraction II of the *M. zeaxanthinifaciens* EtOAc extract (FII) in murine erythrocytes after 2 h of incubation. CT−—negative control/PBS; and CT+—positive control/water. The absorbance was measured in a plate spectrophotometer (λ = 570 nm) (FilterMax F5, Molecular Devices, Molecular Devices, San Jose, CA USA). *** *p* < 0.005.

**Figure 6 pharmaceuticals-17-00499-f006:**
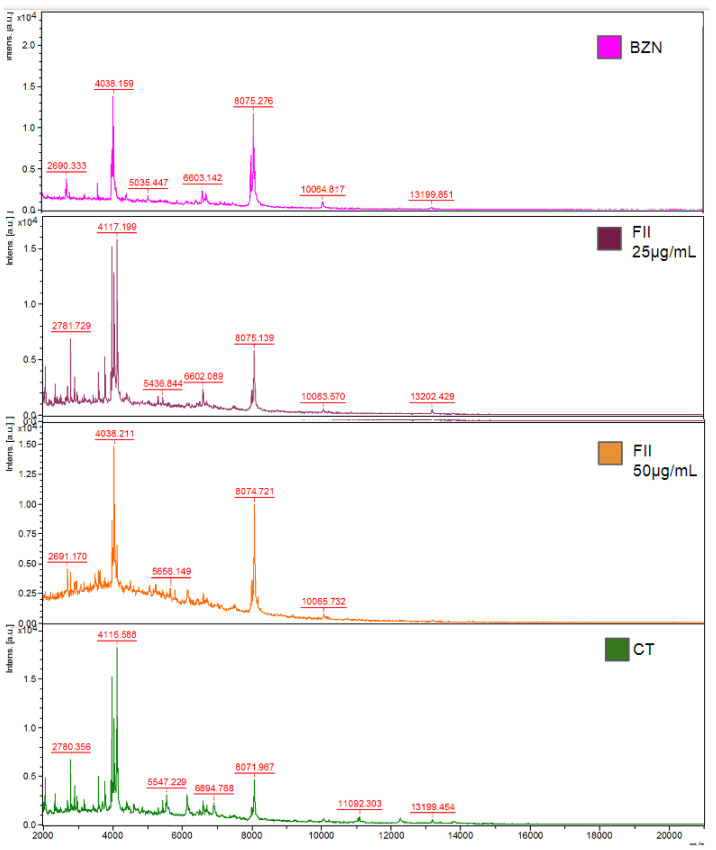
Evaluation of the protein profile of the trypomastigotes of *T. cruzi* in the presence of fraction II from the *M. zeaxanthinifaciens* EtOAc extract (FII) with the MALDI-TOF/MS Microflex. Spectrum of the mass of trypomastigotes treated with FII (25 and 50 μg/mL), treated with benznidazole (BZN-40 μM), and the untreated control (CT). The parasites were treated for 24 h, and the protein profile was compared to the untreated parasites.

**Figure 7 pharmaceuticals-17-00499-f007:**
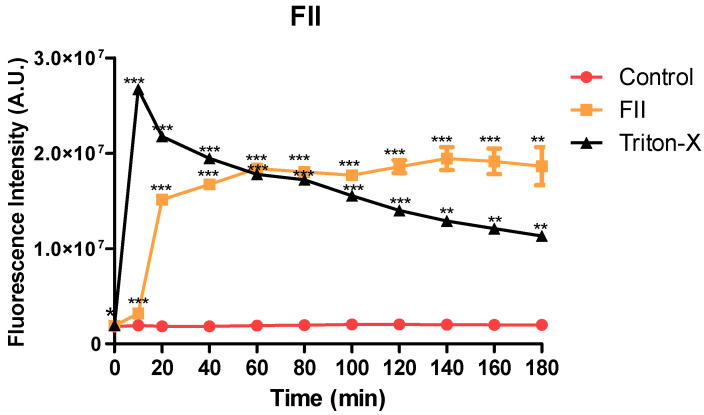
Evaluation of the membrane permeability of the trypomastigotes after treatment with fraction II from the *M. zeaxanthinifaciens* EtOAc extract (FII), using the SYTOX Green dye probe and a spectrofluorometer. The untreated parasites (the control group) and those treated with 0.5% (*v*/*v*) of Triton X-100 were evaluated as the minimal and maximal permeabilization, respectively. *** *p* < 0.0001; and ** *p*< 0.001, * *p* < 0.05.

**Figure 8 pharmaceuticals-17-00499-f008:**
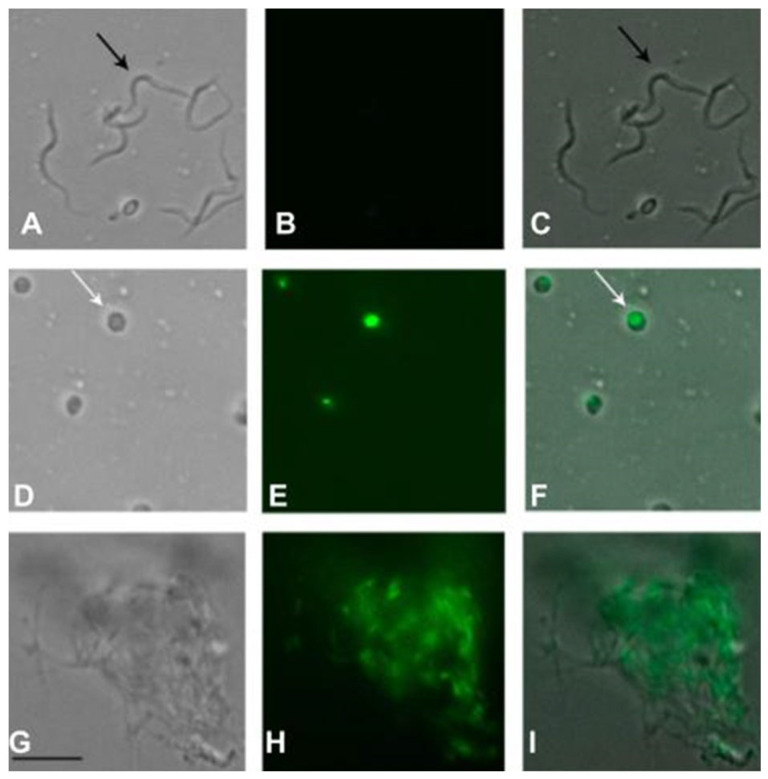
Digital fluorescence images (100× magnification) obtained after treatment of the trypomastigotes with fraction II of the *M. zeaxanthinifaciens* EtOAc extract (FII) (**D**–**F**). The black arrows represent elongated forms of the *T. cruzi* (trypomastigotes) and white arrows represent rounded forms (amastigote-like forms) after treatment. The SYTOX Green dye was used to evaluate the permeabilization effect. The untreated parasites were used as a negative control (**A**–**C**) and the parasites treated with 0.5% (*v*/*v*) of Triton X-100 were used as a positive control for maximal permeabilization (**G**–**I**). A representative experiment is shown. Scale bar 15 µm.

**Table 1 pharmaceuticals-17-00499-t001:** Origin of the microorganisms associated with corals or sediments.

Acronym	Origin	Depth (Meters)
TC	*Tubastraea coccinea*	5
MH	*Mussismilia hispida*	10
MD	*Madracis decactis*	10
SCSB	Sediment of the São Sabastião Channel	35
SIBUZ	Sediment of Buzios Island	13

**Table 2 pharmaceuticals-17-00499-t002:** Identification of bacteria isolated from corals and marine sediments according to MALDI-TOF/MS score or gene sequencing.

Acronym	Identification	Method
TC 2.0.2	*Alteromonas macleodi*	S
TC 2.2	*Vibrio alginolyticus*	MS
MH 3.0	*Vibrio harveyi*	MS
MH 3.3	*Vibrio alginolyticus*	MS
MD 5.0	*Vibrio harveyi*	MS
SCSB 6.0.2.1	*Shewanella pneumatophori*	S
SCSB 6.0.2.2	*Mesoflavibacter zeaxanthinifaciens*	S
SCSB 6.1	*Bacillus megaterium*	MS
SCSB 6.2	*Vibrio harveyi*	MS
SIBUZ 7	*Vibrio harveyi*	MS
SIBUZ 7.2.2	*Halomonas aquamarine*	S

MS—identification by score obtained in MALDI-TOF/MS from two independent studies; S—identification by genetic sequencing; TC—*Tubastraea coccinea*; MH—*Mussismilia hispida*; MD—*Madracis decactis*; SCSB—Sediments of the São Sebastião Channel; and SIBUZ—Sediments of Buzios Island.

**Table 3 pharmaceuticals-17-00499-t003:** Identification of marine strains by partial sequencing of the 16S rRNA gene, according to the BLAST 16S rRNA sequences database.

Acronym	Microorganism	Identity (%)	Query Cover (%)	Acc Number
TC 2.0.2	*Alteromonas macleodii*	99.78	99	OP163900
SCSB 6.0.2.1	*Shewanella pneumatophori*	99.72	99	OP163959
SCSB 6.0.2.2	*Mesoflavibacter zeaxanthinifaciens*	100	96	OR479885
SIBUZ 7.2.2	*Halomonas aquamarina*	99.72	100	OP163958

Acc number: Accession number in GenBank (NCBI).

**Table 4 pharmaceuticals-17-00499-t004:** Quantification of the 50% inhibitory concentration (IC_50_) of microbial metabolites against *T. cruzi* trypomastigotes, with the respective yields of extracted metabolites.

Acronym	Strain	Mass of Metabolites (mg)	IC_50_ ± SD (μg/mL)
TC 2.0.2	*Alteromonas macleodi*	2.6	31.2 ± 2.6
TC 2.2	*Vibrio alginolyticus*	2.1	13.9 ± 3.0
MH 3.0	*Vibrio harveyi*	11.4	51.3 ± 1.1
MH 3.3	*Vibrio alginolyticus*	8.4	25.4 ± 1.2
MD 5.0	*Vibrio harveyi*	3.4	18.3 ± 2.6
SCSB 6.0.2.1	*Shewanella pneumatophori*	5.5	15.1 ± 3.3
SCSB 6.0.2.2	*Mesoflavibacter zeaxanthinifaciens*	4.8	17.9 ± 0.7
SCSB 6.1	*Bacillus megaterium*	2.7	59.9 ± 0.1
SCSB 6.2	*Vibrio harveyi*	3.9	32.8 ± 3.9
SIBUZ 7	*Vibrio harveyi*	8.9	8.0 ± 0.7
SIBUZ 7.2.2	*Halomonas aquamarina*	3.2	15.4 ± 0.5

TC—*Tubastraea coccinea*; MH—*Mussismilia hispida*; MD—*Madracis decactis*; SCSB—Sediments of the São Sebastião Channe; SIBUZ—Sediments of Buzios Island; and SD—Standard Deviation. Benznidazole was used as the standard drug (the positive control).

**Table 5 pharmaceuticals-17-00499-t005:** Evaluation of the 50% Inhibitory Concentration (IC_50_) against the trypomastigotes and amastigotes of *T. cruzi* and the 50% Cytotoxic Concentration (CC_50_) of fraction II from the *M. zeaxanthinifaciens* EtOAc extract.

Compound	Trypomastigotes (IC_50_ ± SD)	Amastigotes (IC_50_ ± SD)	Cytotoxicity (CC_50_ ± SD)
Fraction II	17.7 ± 3.5 μg/mL	23.8 ± 2.7 μg/mL	>200 μg/mL
Benznidazole	3.6 ± 0.9 μg/mL	1.4 ± 0.5 μg/mL	49.4 μg/mL

SD—standard deviation; IC_50_—50% Inhibitory Concentration; and CC_50_—50% Cytotoxic Concentration.

## Data Availability

The 16S rRNA data presented in this study are openly available in GeneBank (NCBI), reference number OP163900, OP163959, OR479885, and OP163958. The raw data supporting the conclusions of this article will be made available by the authors on request.

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
