# Peer review of "Saturated Iso-Type Fatty Acids from the Marine Bacterium Mesoflavibacter zeaxanthinifaciens with Anti-Trypanosomal Potential"

_pharmaceuticals, 2024, doi:10.3390/ph17040499_

Round 1

Reviewer 1 Report

Comments and Suggestions for Authors

The manuscript of Santos Ferreira et al. focusses in the isolation of different bacteria from different marine samples and the study of the effect of the ethanolic extracts against Trypanosoma cruzi. A deeper study is done with a fraction isolated from Mesoflavibacter zeaxanthinifaciens on which they identified iso-type fatty acids as possible effectors of the lethal effect observed in trypomastigotes and amastigotes.

My comments

Please revise all the text (references included) to use italics in the name of microorganisms. The complete name of the organism must appear complete the first time that is mentioned and after that, the genus name should be abbreviated.

Fig 1, please use a more visible color for the arrow that indicates the location and include the coordinates in some part of the text (Now they are included in the materials and methods part).

Table 2, Bacillus megaterium is G+ and not G- as appears in the table.

Table 2 only shows 11 microorganisms included in 7 species. Are these the only species obtained in the plates or are these just the selected ones maybe due to their abundance?

For the reader it is not clear the reason for selecting the strain M. zeaxanthinifaciens to deepen in the study the produced compounds. Other of the isolated strains look more active. Have the authors fractionated the extracts obtained from these other strains?

L145-150 The authors stated “Based on the dose-response sigmoidal curves (data not shown), six microbial extracts killed 100% of the parasites to the highest tested………….”.  The inclusion of a supplementary fig with these data may improve the understanding of this part.

L156-179. The authors identified 4 iso-type fatty acids in the fraction II of M. zeaxanthinifaciens as the active molecules with killing potential against trypomastigotes and amastigotes. A deeper study will allow to separate these compounds and to identify the real active molecule. Are some of these compounds commercial in order to study the individual effect of them?

Fig. 5. Please indicate what is the positive control used in this study in some part of the text.  What is the hemolytic activity of benznidazole?

L198-99 I think that the sentence “No hemolysis could …………” is incorrect, please correct.

L210-227 The authors state “based I the spectra data…………”. However, some of the peaks mentioned in the text are not indicated in the figure 6. Please include all the peaks indicated in the text in the Figure 6.

L 220.  The peak referred as 4.417 is nor right it must be 4.117

L230-236, please indicate the concentration of the fraction II and triton X100 used.

Fig 8, in the panels G, H, I it is not possible to see the parasite. Is it possible to include better panels?

L289, Streptomyces griseus is a species and not a genus as it is indicated in the text, please correct.

L333 Please use the complete name for S. aureus (it is the first time that it appears in the text).

L 453, please indicate the concentration of the RNase used.

 L466-471, The authors analyze the active compounds present in the cells. Have the authors considered to study the secreted compounds present in the agar after the culture?

L467 Please indicate the volume of water used.

L488 Please corroborate that the indication 9:13:2 is right (only methanol and water are used).

Author Response

Referee 1.

Referee 1: The manuscript of Santos Ferreira et al. focusses in the isolation of different bacteria from different marine samples and the study of the effect of the ethanolic extracts against Trypanosoma cruzi. A deeper study is done with a fraction isolated from Mesoflavibacter zeaxanthinifaciens on which they identified iso-type fatty acids as possible effectors of the lethal effect observed in trypomastigotes and amastigotes.

Please revise all the text (references included) to use italics in the name of microorganisms. The complete name of the organism must appear complete the first time that is mentioned and after that, the genus name should be abbreviated.

Author. Thank you for the comments. The text was revised according to your suggestion and the modifications were highlighted in yellow.

Referee 1: Fig 1, please use a more visible color for the arrow that indicates the location and include the coordinates in some part of the text (Now they are included in the materials and methods part).

Author. Thank you for the suggestion. The Figure 1 was improved and the coordinates were included, accordingly.

Referee 1: Table 2, Bacillus megaterium is G+ and not G- as appears in the table.

Author. Thank you for the suggestion. We removed the column with this information as we concluded that it was not relevant.

Referee 1: Table 2 only shows 11 microorganisms included in 7 species. Are these the only species obtained in the plates or are these just the selected ones maybe due to their abundance?

Author. Thank you for your comments. We isolation procedure yielded 11 microorganisms and they were selected based in their abundance. This information is now included in the text as bellow:

“2.2. Isolation and Identification of Marine Bacteria. The microbiota associated to the corals was isolated under sterile conditions and stored at -85°C. We isolation procedure yielded 11 microorganisms and they were selected based in their abundance.”

Referee 1: For the reader it is not clear the reason for selecting the strain M. zeaxanthinifaciens to deepen in the study the produced compounds. Other of the isolated strains look more active. Have the authors fractionated the extracts obtained from these other strains?

Author. Thank you for your comments. We did not fractionate the extracts obtained from other strains. In the Discussion, we improved the paragraph describing the potential of M. zeaxanthinifaciens regarding bioactivities published in the literature. Despite the metabolites from Vibrio harveyi were the most potent against T. cruzi, this specie has been widely explored in the literature for different biological activities with some isolated compounds. The selection of M. zeaxanthinifaciens was based in the novelty for the antiparasitic potential and this specie also showed a good relation between the IC50 value and yield of metabolites (mass) among other isolated strains. Additionally, to the best of our knowledge, we presented for the first time, the isolation of M. zeaxanthinifaciens in the Atlantic Ocean, as it was previously described in Japan (Pacific Ocean). According to your comments, we included this information in the Discussion.

Referee 1: L145-150 The authors stated “Based on the dose-response sigmoidal curves (data not shown), six microbial extracts killed 100% of the parasites to the highest tested………….”.  The inclusion of a supplementary fig with these data may improve the understanding of this part.

Author. Thank you for your suggestion. We included the Fig S1, with the information “Dose-response curves of microbial metabolites against T. cruzi trypomastigotes. TC (Tubastraea coccinea), MH (Mussismilia hispida), MD (Madracis decactis), SCSB (Sediments of the São Sebastião Channel), SIBUZ (Sediments of Buzios Island).”

Referee 1: L156-179. The authors identified 4 iso-type fatty acids in the fraction II of M. zeaxanthinifaciens as the active molecules with killing potential against trypomastigotes and amastigotes. A deeper study will allow to separate these compounds and to identify the real active molecule. Are some of these compounds commercial in order to study the individual effect of them?

Author. Thank you for your comments. As reported in the manuscript, the bioactive fraction was shown to be composed of a mixture of iso-fatty acids which were characterized by analysis of NMR and MS data. However, the reduced amount and the complexity of this fraction did not allow the conduction of additional chromatographic methods aiming purification of each identified acid. The isolation of these homologous compounds is a difficult task because the purification must be conducted only by HPLC – additionally, the absence of a chromophore in the molecular structure restricts the use of common detectors such as UV. Based on these points, we decided to evaluate the antitrypanosomal effect of this mixture of chemically related iso-fatty acids instead of testing the separated compounds. In the literature, long-chain fatty acids, isolated from terrestrial sources, have been demonstrated antimalarial, antimycobacterial and antifungal properties [Carballeira, 2008]. According to Casillas-Vargas et. al. (2021) antibacterial activity of fatty acids is well known in the literature and represents a promising option for developing the next-generation of antibacterial agents. It includes the activity of saturated fatty acids as lauric acid and others, with properties against Gram-positive and Gram-negative bacteria [Yang et al, 2017, Huang et al, 2014, Nitbani et al, 2016). We found no commercial iso-fatty acids with the same formula of our FII fraction.

References

Carballeira, N.M. New Advances in Fatty Acids as Antimalarial, Antimycobacterial and Antifungal Agents. Prog Lipid Res 2008, 47, 50–61, doi:10.1016/j.plipres.2007.10.002.

Casillas-Vargas G, Ocasio-Malavé C, Medina S, Morales-Guzmán C, Del Valle RG, Carballeira NM, Sanabria-Ríos DJ. Antibacterial fatty acids: An update of possible mechanisms of action and implications in the development of the next-generation of antibacterial agents. Prog Lipid Res. 2021 Apr;82:101093. doi: 10.1016/j.plipres.2021.101093. Epub 2021 Feb 9. PMID: 33577909; PMCID: PMC8137538.

Yang H, Chen J, Rathod J, Jiang Y, Tsai P, Hung Y, et al. Lauric acid is an inhibitor of Clostridium difficile growth in vitro and reduces inflammation in a mouse infection model. Front Microbiol 2017;8:2635. [PubMed: 29387044]

Huang W, Tsai T, Chuang L, Li YY, Zouboulis C, Tsai P. Anti-bacterial and anti-inflammatory properties of capric acid against Propionibacterium acnes: A comparative study with lauric acid. J Dermatol Sci 2014;73(3):232–40. [PubMed: 24284257]

Nitbani F, Jumina J, Siswanta D, Sholikhah E. Isolation and antibacterial activity test of lauric acid from crude coconut oil (Cocos nucifera L.). Procedia Chem 2016; 18:132–40

Referee 1: Fig. 5. Please indicate what is the positive control used in this study in some part of the text.  What is the hemolytic activity of benznidazole?

Author. Thank you for your comments. We included in the Methods item 4.1.2 and at Results (item 2.6) the following : “Water was used to induce 100% of hemolysis (positive control).” Benznidazole is the standard drug in clinical use and usually, no hemolytic activity is performed for approved-drugs. Water was the internal control.

Referee 1: L198-99 I think that the sentence “No hemolysis could …………” is incorrect, please correct.

Author. Thank you for the correction. We modified to “No hemolytic activity could…”, accordingly.

Referee 1: L210-227 The authors state “based I the spectra data…………”. However, some of the peaks mentioned in the text are not indicated in the figure 6. Please include all the peaks indicated in the text in the Figure 6.

Author. Thank you for the comments. As requested, we included a table at the supplementary material for better comparisons as bellow:

Table S1- Protein signals (m/z) in MALDI-TOF/MS of trypomastigotes of T. cruzi in the presence of fraction II from M. zeaxanthinifaciens EtOAc extract (FII). Trypomastigotes were treated with FII at 25 and 50 μg/mL, and benznidazole (BZN-40 μM) was used as standard drug. Parasites without treatment were used as control (untreated). The parasites were treated for 24 h and the protein profile was compared to untreated parasites.”

Referee 1: L 220.  The peak referred as 4.417 is nor right it must be 4.117

Author. Thank you for the comments. As requested, we corrected it and included a table at the supplementary material for better comparisons (Table S1).

Referee 1: L230-236, please indicate the concentration of the fraction II and triton X100 used.

Author. We included the concentration of FII (17.7 µg/mL) and Triton X-100 (0.5% v/v) as requested.

Referee 1: Fig 8, in the panels G, H, I it is not possible to see the parasite. Is it possible to include better panels?

Author. Thank you for the comments. After incubation with Triton X-100, the parasite is permeabilized and the cells lose their morphology. Using Sytox Green, this dye enters the cell after permeabilization and binds to nucleic acids, enhancing 500-fold its fluorescence levels. The images G,H,I are at the highest resolution and due to this deleterious effect of the detergent in the cells, the images show damaged cells with a completely altered morphology.

Referee 1: L289, Streptomyces griseus is a species and not a genus as it is indicated in the text, please correct.

Author. Thank you for the comments. It was corrected accordingly.

Referee 1: L333 Please use the complete name for S. aureus (it is the first time that it appears in the text).

Author. Thank you for the comments. It was corrected accordingly.

Referee 1: L 453, please indicate the concentration of the RNase used.

Author. Thank you for the comments. We indicated the concentration of the RNase used, accordingly.

Referee 1: L466-471, The authors analyze the active compounds present in the cells. Have the authors considered to study the secreted compounds present in the agar after the culture?

Author. Thank you for the comments. Indeed, secreted compounds present in the agar after cultivation of the bacteria, may have potential compounds with antiparasitic activity. Unfortunately, the medium (Marine Broth, with agar) presented anti-Trypanosoma cruzi activity in our previous studies. To avoid misleading, we opted to carefully scrap the bacteria from the agar, and no organic extracts of the remaining agar were prepared.

Referee 1: L467 Please indicate the volume of water used.

Author. As requested, we included the volume of water (200 mL for every 20 plates).

Referee 1: L488 Please corroborate that the indication 9:13:2 is right (only methanol and water are used).

Author. As requested, we corrected the text as below:

“…MeOH:H2O (1:9, 3:7, 6:4 v/v and pure MeOH, respectively) as eluents”

Reviewer 2 Report

Comments and Suggestions for Authors

After carefully reading the manuscript (Pharmaceuticals-2901792) entitled “Unsaturated Iso-Type Fatty Acids from the Marine Bacterium Mesoflavibacter zeaxanthinifaciens with  Antitrypanosomal Potential” by Ferreira et al. The manuscript is well written and its content of figures, analysis, and tables is very sufficient. I recommend accepting this manuscript for publication in a Pharmaceuticals journal after a minor modification in the English language and review with English native speakers because there are a few spelling, grammatical, and abbreviation errors.

Comments on the Quality of English Language

Minor

Author Response

Referee 2

Referee 2: After carefully reading the manuscript (Pharmaceuticals-2901792) entitled “Unsaturated Iso-Type Fatty Acids from the Marine Bacterium Mesoflavibacter zeaxanthinifaciens with  Antitrypanosomal Potential” by Ferreira et al. The manuscript is well written and its content of figures, analysis, and tables is very sufficient. I recommend accepting this manuscript for publication in a Pharmaceuticals journal after a minor modification in the English language and review with English native speakers because there are a few spelling, grammatical, and abbreviation errors.

Author. Thank you for the comments. Our co-author, Prof. Myron Christodoulides from University of Southampton (UK) is a native English speaker, and corrected the final version of the revised manuscript.

Reviewer 3 Report

Comments and Suggestions for Authors

Comments

The paper entitled “Unsaturated Iso-Type Fatty Acids from the Marine Bacterium Mesoflavibacter zeaxanthinifaciens with Antitrypanosomal Potential” written by Dayana Agnes Santos Ferreira and co-authors described the antiparasitic potential of bacterial metabolites isolated from three corals and sediments collected from the North coast of São Paulo, Brazil. Chagas disease is a Neglected Tropical Disease with limited and ineffective therapy. The current treatment employs only two highly toxic drugs, benznidazole or nifurtimox, but both associate with severe adverse effects. Thus, the development of safer and more effective drugs for Chagas disease is urgently needed. These findings demonstrate unsaturated iso-type fatty acids as possible new hits against T. cruzi. Generally, this work meets the requirement of this journal. However, there are some issues in the paper that need to be carefully addressed. Above all, I recommend this paper to be published after major revision.

Some points:

1. Line 69, to our knowledge, approximately 40,000 marine natural products have been reported to date according to the marinLit database (https://marinlit.rsc.org/). It is strongly suggested to check the data by literatures.

2. As the authors noted that the most active extracts were from Vibrio spp. with IC50 values ranging from 8 to 51 μg/mL (Line 142).  It is confused that Mesoflavibacter zeaxanthinifaciens was further subjected to evaluation of its fractions against trypomastigotes. As shown in table 4, the potency (IC50= 18 μg/mL) of Mesoflavibacter zeaxanthinifaciens along with the amount (4.8 g) of its EtOAc extracts revealed it is not the representative material. Otherwise, Vibrio harveyi (EtOAc extracts: 8.9 g, IC50= 8.0 μg/mL) is the best investigated material. Thus, this part should be discussed in detail.

3. The EtOAc extracts of Mesoflavibacter zeaxanthinifaciens showed moderate inhibition against T. cruzi trypomastigotes with an IC50 value of 18 μg/mL. Meanwhile, the bioactive fraction II of Mesoflavibacter zeaxanthinifaciens EtOAc extracts also showed equivalent inhibition against T. cruzi trypomastigotes (17.7 μg/mL). Generally, the potency of purified fraction II of Mesoflavibacter zeaxanthinifaciens should be stronger than its crude extracts. This phenomenon should also be further discussed in depth.

4. Four kinds of iso-type fatty acids were identified as bioactive constituents from the fraction II of Mesoflavibacter zeaxanthinifaciens EtOAc extracts by NMR and MS methods. Generally, fatty acids are not bioactive constituents in the point view of natural product chemistry. Besides, as shown in the above-mentioned issue, the fraction II of Mesoflavibacter zeaxanthinifaciens EtOAc extracts didn’t show stronger activity. Thus, in order to corroborate the bioactive constituents of fatty acids, it is strongly suggested to evaluated the anti-T. cruzi trypomastigotes of a pure iso-type fatty acid by chromatographic resolution.

5. In figure 5, the labels of significance should be improved.

6. Some trivials:

Ø  All the latin names should be in italic types, and please check the whole manuscript carefully. Lines 95, 108, 117, 118, 124, 128, 135, 139-148, 153, 166, 185, 335, etc.

Ø  Please check the spelling mistakes in the whole manuscript. Lines 486, 508, 510, 549,

Ø  italic type v/v---v/v, m/z---m/z

Ø  the NMR spectra should be provided in the supporting information.

Author Response

Referee 3

Referee 3 Line 69, to our knowledge, approximately 40,000 marine natural products have been reported to date according to the marinLit database (https://marinlit.rsc.org/). It is strongly suggested to check the data by literatures.

Author. Thank you for the correction. We included the following text: The marine environment has been a source of more than 40,000 compounds from marine macro and microorganisms, accordingly to MarinLit database [14].”

Referee 3: a-  As the authors noted that the most active extracts were from Vibrio spp. with IC50 values ranging from 8 to 51 μg/mL (Line 142).  It is confused that Mesoflavibacter zeaxanthinifaciens was further subjected to evaluation of its fractions against trypomastigotes. As shown in table 4, the potency (IC50= 18 μg/mL) of Mesoflavibacter zeaxanthinifaciens along with the amount (4.8 g) of its EtOAc extracts revealed it is not the representative material. Otherwise, Vibrio harveyi (EtOAc extracts: 8.9 g, IC50= 8.0 μg/mL) is the best investigated material. Thus, this part should be discussed in detail.

Author. Thank you for your comments. In the Discussion, we improved the paragraph describing the potential of M. zeaxanthinifaciens regarding bioactivities published in the literature. Despite the metabolites from Vibrio harveyi were the most potent against T. cruzi, this specie has been widely explored in the literature for different biological activities with some isolated compounds. The selection of M. zeaxanthinifaciens was based in the novelty for the antiparasitic potential and this specie also showed a good relation between the IC50 value and yield of metabolites (mass) among the isolated strains. Additionally, to the best of our knowledge, we presented for the first time, the isolation of M. zeaxanthinifaciens in the Atlantic Ocean, as it was previously described in Japan (Pacific Ocean). According to your comments, we included this information in the Discussion.

Referee 3: The EtOAc extracts of Mesoflavibacter zeaxanthinifaciens showed moderate inhibition against T. cruzi trypomastigotes with an IC50 value of 18 μg/mL. Meanwhile, the bioactive fraction II of Mesoflavibacter zeaxanthinifaciens EtOAc extracts also showed equivalent inhibition against T. cruzi trypomastigotes (17.7 μg/mL). Generally, the potency of purified fraction II of Mesoflavibacter zeaxanthinifaciens should be stronger than its crude extracts. This phenomenon should also be further discussed in depth.

Author. Thank you for your comments. Indeed, fractionated samples are usually more active than the original crude extract, because the fractionation procedure itself, concentrates the active compounds. According to your suggestion, we included the following paragraph in the Discussion:

Additionally, to the best of our knowledge, we presented for the first time, the isolation of M. zeaxanthinifaciens in the Atlantic Ocean, as it was previously described in Japan (Pa-cific Ocean) [18]. Using solid-phase extraction, we conducted a fractionation of the crude EtOAc extract from M. zeaxanthinifaciens, resulting in FII, a fraction with trypanocidal effect similar to the crude extract, as confirmed by the lack of mitochondrial activity, detected by the resazurin assay. Usually, fractionated samples are more active than the original crude extract, because the fractionation procedure itself, concentrates the active compounds. In our study, FII is not the most abundant fraction, corresponding to 2.9% of the total mass of crude extract and this fact, cannot explain the similarity between the IC50 values. Considering that synergism between compounds in crude extracts has been widely reported (Vidar et al., 2023), the separation of compounds in FII from those in F0, FI, FIII, may have contributed to reduce the potency against T. cruzi parasites.

Reference

Vidar WS, Baumeister TUH, Caesar LK, Kellogg JJ, Todd DA, Linington RG, M Kvalheim O, Cech NB. Interaction Metabolomics to Discover Synergists in Natural Product Mixtures. J Nat Prod. 2023 Apr 28;86(4):655-671. doi: 10.1021/acs.jnatprod.2c00518. Epub 2023 Apr 13. PMID: 37052585; PMCID: PMC10152448.

Referee 3: c-  Four kinds of iso-type fatty acids were identified as bioactive constituents from fraction II of Mesoflavibacter zeaxanthinifaciens EtOAc extracts by NMR and MS methods. Generally, fatty acids are not bioactive constituents in the point view of natural product chemistry. Besides, as shown in the above-mentioned issue, the fraction II of Mesoflavibacter zeaxanthinifaciens EtOAc extracts didn’t show stronger activity. Thus, in order to corroborate the bioactive constituents of fatty acids, it is strongly suggested to evaluated the anti-T. cruzi trypomastigotes of a pure iso-type fatty acid by chromatographic resolution.

Author. Thank you very much for your comments, it makes improvements in our Discussion. We found biological activities for fatty acids in the literature. According to Casillas-Vargas et. al. (2021 -reference bellow), antibacterial activity of fatty acids is well known in the literature and represents a promising option for developing the next-generation of antibacterial agents. It includes the activity of saturated fatty acids as lauric acid and others, with properties against Gram-positive and Gram-negative bacteria [Yang et al, 2017, Huang et al, 2014, Nitbani et al, 2016).

Regarding the isolation, in fact, the purification of each iso-type fatty acid was tentatively performed by RP-HPLC (using both C8 and C18 columns). However, due to the absence of a chromophore in the structures of these chemically related compounds, it was not possible the detection using UV – even using DAD in our equipment. Therefore, considering the possible synergistic effect of these compounds, we decide to evaluate the antitrypanosomal activity of fraction II in this work and, in future studies, we will try to synthesize them using as starting material w 1-unsaturated fatty acids, previously isolated from different plant species such as Porcelia macrocarpa (Brito et al., 2021).

References

Brito, Ivanildo A.; Oliveira, Emerson A.; Chaves, Mariana H.; Thevenard, Fernanda; Rodrigues-Oliveira, André F.; Barbosa-Reis, Gustavo; Sartorelli, Patricia; Oliveira-Silva, Diogo; Tempone, Andre G.; Costa-Silva, Thais A.; Lago, João Henrique G. Antileishmanial Acetylene Fatty Acid and Acetogenins from Seeds of Porcelia macrocarpa. J. Braz. Chem. Soc., 32, 447-453, 2021.

Casillas-Vargas G, Ocasio-Malavé C, Medina S, Morales-Guzmán C, Del Valle RG, Carballeira NM, Sanabria-Ríos DJ. Antibacterial fatty acids: An update of possible mechanisms of action and implications in the development of the next-generation of antibacterial agents. Prog Lipid Res. 2021 Apr;82:101093. doi: 10.1016/j.plipres.2021.101093. Epub 2021 Feb 9. PMID: 33577909; PMCID: PMC8137538.

Yang H, Chen J, Rathod J, Jiang Y, Tsai P, Hung Y, et al. Lauric acid is an inhibitor of Clostridium difficile growth in vitro and reduces inflammation in a mouse infection model. Front Microbiol 2017;8:2635. [PubMed: 29387044]

Huang W, Tsai T, Chuang L, Li YY, Zouboulis C, Tsai P. Anti-bacterial and anti-inflammatory properties of capric acid against Propionibacterium acnes: A comparative study with lauric acid. J Dermatol Sci 2014;73(3):232–40. [PubMed: 24284257]

Nitbani F, Jumina J, Siswanta D, Sholikhah E. Isolation and antibacterial activity test of lauric acid from crude coconut oil (Cocos nucifera L.). Procedia Chem 2016; 18:132–40

Referee 3 In figure the 5, labels of significance should be improved.

Author. Thank you for your correction. We included improved it accordingly.

Referee 3

Some trivials:

Ø  All the latin names should be in italic types, and please check the whole manuscript carefully. Lines 95, 108, 117, 118, 124, 128, 135, 139-148, 153, 166, 185, 335, etc.

Author. Thank you for your correction. We corrected it accordingly.

Referee 3

Ø  Please check the spelling mistakes in the whole manuscript. Lines 486, 508, 510, 549,

Author. Thank you for your correction. We corrected it accordingly.

Referee 3

Ø  italic type: v/v---v/v, m/z---m/z

Author. Thank you for your correction. We corrected it accordingly.

Referee 3

Ø  the NMR spectra should be provided in the supporting information.

Author. Thank you for your correction. We included the NMR spectra in the supporting information.

Reviewer 4 Report

Comments and Suggestions for Authors

This study isolated several marine bacteria extracts and characterized their antitrypanosomal properties.

In general, the manuscript is well logically organized with enough experimental data to support the major conclusion. The outcome is also useful in this fiend as well.

Major issues:

1)      Detailed pharmaceutical effects of iso-type fatty acids which were mentioned in this study should be further studied their antitrypanosomal effects. Logically, each purified individual STFS should be given their data. The can be done by supplemented experiments or via clear reference support.

2)      Lack of in-depth discussion why STFSs have antitrypanosomal effects, only altering the permeability of the membrane is not a convincing evidence, any other more specific effects?

3)      Marine Bacterium extracts can be directly utilized in clinical, is their any practical application in market? Add more discussion and references.

Minor issues:

1)      Latin name should be present in italic form. Line 138-150, Line 166, line 185, 208,231…..

2)      The quality and resolution of fig.1, fig.5, fig.8 should be improved.

Author Response

Referee 4

Referee 4 -     This study isolated several marine bacteria extracts and characterized their antitrypanosomal properties. In general, the manuscript is well logically organized with enough experimental data to support the major conclusion. The outcome is also useful in this fiend as well.

Major issues:

Referee 4 -     Detailed pharmaceutical effects of iso-type fatty acids which were mentioned in this study should be further studied their antitrypanosomal effects. Logically, each purified individual STFS should be given their data. The can be done by supplemented experiments or via clear reference support.

Author. Thank you for your comments. Initially, we would like to inform you that the purification of each iso-type fatty acid was tentatively performed by RP-HPLC (using both C8 and C18 columns). However, due to the absence of a chromophore in the structures of these chemically related compounds, it was not possible the detection using UV – even using DAD in our equipment. Therefore, considering the possible synergistic effect of these compounds, we decide to evaluate the anti-trypanosomal activity of fraction II in this work. The effect of different fatty acids against T. cruzi was previously reported in the literature (Santos et al., Molecules, 2015, 20, 8168-8180; Kaneda et al., Microbiological Reviews, 1991, 288-302) – as observed, chemically related compounds affect the parasite causing plasma membrane disruption (Londero et al., Bioorganic Chemistry, 2018, 78, 307–311). Due to the purification of these compounds being difficult, we intend to synthesize related iso-acids using as starting material w1-unsaturated derivatives, previously isolated from different plant species such as Porcelia macrocarpa (Brito et al. 2021) aiming to perform mechanism of action (MoA) studies to investigate how these compounds affect the parasite.

References

Londero VS, da Costa-Silva TA, Gomes KS, Ferreira DD, Mesquita JT, Tempone AG, Young MCM, Jerz G, Lago JHG. Acetylenic fatty acids from Porcelia macrocarpa (Annonaceae) against trypomastigotes of Trypanosoma cruzi: Effect of octadec-9-ynoic acid in plasma membrane electric potential. Bioorg Chem. 2018 Aug;78:307-311. doi: 10.1016/j.bioorg.2018.03.025. Epub 2018 Mar 30. PMID: 29625270.

Brito, Ivanildo A.; Oliveira, Emerson A.; Chaves, Mariana H.; Thevenard, Fernanda; Rodrigues-Oliveira, André F.; Barbosa-Reis, Gustavo; Sartorelli, Patricia; Oliveira-Silva, Diogo; Tempone, Andre G.; Costa-Silva, Thais A.; Lago, João Henrique G. Antileishmanial Acetylene Fatty Acid and Acetogenins from Seeds of Porcelia macrocarpa. J. Braz. Chem. Soc., 32, 447-453, 2021.

Referee 4-      Lack of in-depth discussion why STFSs have antitrypanosomal effects, only altering the permeability of the membrane is not a convincing evidence, any other more specific effects?

Author. Thank you for your comments. Alteration of the plasma membrane, if not reversed, leads to cell leakage and lost of ions, culminating in cell death. Our microscopy data presented in Figure 8 (C,D,E) clearly shows a complete altered morphology of the parasite to a round shape, with penetration of the vital fluorescent probe SYTOX Green. Additionally, using the resazurin assay to determine the IC50 values, we could also observe the lack of mitochondrial activity, compatible to a cellular death.

Usually, we perform no additional mechanism of action (MoA) studies after a SYTOX Green-positive assay. Mechanism of action studies are usually performed at short-time incubation with the testing compound to evaluate the initial damages in the cell. If no alteration of plasma membrane permeability of T. cruzi is detected at this point, further studies are needed to investigate the target organelles. But in the present study, using the fluorescent probe SYTOX Green, we could detect a rapid (10 minutes) increase of the fluorescent levels (by spectrofluorimetric assay Figure 7) after incubation with FII, corresponding to the alteration of the plasma membrane permeability. According to the probe’s manufacturer (Thermo), after penetration of SYTOX in damaged membranes, it binds to DNA and enhance 500-fold its fluorescence. This effect could also be observed by fluorescence microscopy (Figure 8), with complete deterioration of the parasite’s morphology to a round shape when compared to untreated parasites (Fig. 8 A,C), compatible to a dead cell. The fluorescence continued increasing to reach a maximal point after 60 minutes. After alteration of the plasma membrane, leakage occurs with extravasation of ions, leading to cell death.

Usually, no additional studies are needed to detect further intracellular damages, that will naturally occur due to the cell leakage. Then, further intracellular damages could not be considered “targets” of the testing compound.

Referee 4-      Marine Bacterium extracts can be directly utilized in clinical, is their any practical application in market? While there are practical applications for marine bacterium extracts, further research and development are needed to fully explore their potential and bring products to market. Additionally, regulatory approval and commercialization processes must be considered for any practical applications in clinical settings.

Author. Thank you for your comments. Indeed. The marine bacterium M. zeaxanthinifaciens, isolated in our study, produces the carotenoid pigment zeaxanthin (3,3′-dihydroxy-β-carotene), which is a yellow lipophilic molecule useful in pigmentation of food products and cosmetics (Zhang et.al, 2017). But to the best of our knowledge, there is no current use of marine bacteria crude extracts in clinical use. But many isolated compounds with significant pharmaceutical potential have been proposed.

Reference

Zhang, Y.; Liu, Z.; Sun, J.; Xue, C.; Mao, X. Biotechnological Production of Zeaxanthin by Microorganisms. 2017, doi:10.1016/j.tifs.2017.11.006.

Referee 4

Minor issues:

1) Latin name should be present in italic form. Line 138-150, Line 166, line 185, 208,231…..

Author. Thank you. We corrected it accordingly.

Referee 4

2) The quality and resolution of fig.1, fig.5, fig.8 should be improved.

Author. Thank you. We improved fig.1, fig.5, fig.8 accordingly and saved them as TIFF images.

Round 2

Reviewer 3 Report

Comments and Suggestions for Authors

This paper has been improved well.